# Factors Associated with Student Satisfaction and Self-Confidence in Simulation Learning among Nursing Students in Korea

**DOI:** 10.3390/healthcare11081060

**Published:** 2023-04-07

**Authors:** Mi-Kyoung Cho, Mi Young Kim

**Affiliations:** 1Department of Nursing Science, Chungbuk National University, 1 Chungdae-ro, Seowon-gu, Cheongju KR28644, Republic of Korea; 2Department of Nursing, College of Nursing, Hanyang University, Seoul KR15588, Republic of Korea

**Keywords:** nursing student, student satisfaction and self-confidence in learning, simulation design scale, educational practices in simulation

## Abstract

This study aimed to examine the relationships between student satisfaction and self-confidence in learning (SCLS), the simulation design scale (SDS), and educational practices in simulation (EPSS) and to identify the influencing factors on SCLS in nursing students undergoing simulation learning. Of the fourth-year nursing students, 71 who were taking a medical–surgical nursing simulation course and voluntarily provided informed consent to participate in the study were enrolled. Data on SCLS, SDS, and EPSS were collected via an online survey after the simulation, from 1 October 2019 to 11 October 2019. The mean SCLS score was 56.31 ± 7.26, the mean SDS score was 86.82 ± 10.19 (range: 64~100), and the mean EPSS score was 70.87 ± 7.66 (range: 53~80). SCLS was positively correlated with SDS (r = 0.74, *p* < 0.001) and EPSS (r = 0.75, *p* < 0.001). The regression model for SCLS in nursing students revealed that SCLS increased with increasing EPSS and SDS, and that SDS and EPSS explained 58.7% of the variance in SCLS (F = 50.83, *p* < 0.001). Therefore, to improve the learning satisfaction and learning confidence of nursing students in simulation classes, it is necessary to consider simulation design and practice considering educational factors.

## 1. Introduction

Simulation-based learning (SBL) in nursing provides a safe environment for clinical practicum in which students can explore nursing scenarios and gain clinical exposure [1] and is utilized in various forms in nursing education [2]. The World Health Organization (WHO) recommends using simulation in education, as it helps students acquire skills and attain improved learning outcomes [3]. In particular, the importance of SBL is further highlighted as an actual clinical practicum owing to the increased barriers to clinical practicum amid a climate focusing on patient safety, human rights, and an infectious disease crisis, namely the coronavirus disease 2019 (COVID-19) pandemic. SBL has been documented to enhance clinical judgment, problem-solving skills, and critical thinking [4] and improve learning satisfaction, confidence, and self-efficacy [5].

However, studies have also reported the negative aspects of simulation, such as unfamiliarity with the novel learning method and students’ anxiety about being evaluated by peers and faculty or making errors [6]. A high level of anxiety among students lowers their confidence in learning, leading to frequent errors and, consequently, further intimidation, which in turn causes them to remain passive during the simulation [7]. As simulation enables students to acquire skills and integrate them into newly learned theories through repeated learning in a safe environment without the risk of causing harm to patients [8], a simulation design that can maximize the positive impact of simulations is essential.

Therefore, the simulation design, elements, and outcomes must be reviewed. Various designs and structures have been experimented with and implemented to minimize the negative aspects of simulations and maximize the positive aspects, such as boosting satisfaction and confidence. One such attempt was made by Jeffries [9], who proposed a simulation framework that laid the foundation for development, implementation, planning, and evaluation. Subsequently, the framework was continually refined, eventually leading to the development of the National League of Nursing (NLN) Jeffries simulation theory. This theory’s simulation frameworks comprise context, background, design, educational practices, simulation experience, and outcomes [10,11]. In other words, the NLN/Jeffries simulation framework was developed to support facilitators of nursing simulations to enhance the quality of SBL [10]. It designates the features to be included in three major aspects of simulation development: simulation design characteristics (objectives, fidelity, problem-solving, student support, and debriefing); educational practices (active learning, feedback, student/faculty interaction, collaboration, high expectations, diverse learning, and time on task); and outcomes (learning, knowledge, skill performance, learner satisfaction, critical thinking, and self-confidence) [12].

Although the simulation design characteristics, educational practices, and outcomes have been theorized, they have not been adequately validated. Existing studies on simulation in Korea have been primarily focused on knowledge and clinical performance [13,14,15], problem-solving skills [14], skills performance, and confidence and satisfaction [14], with a lack of studies that explored the association between simulation education context and design features proposed by Jeffries [9]. In other words, SBL can utilize diverse scenarios, and although SBL using human patient simulators is anticipated to have complex clinical relevance that may be directly linked to patients’ lives, studies assessing simulation design characteristics, educational practices, and empirical validation of their effects on the outcomes are lacking.

Therefore, this study aims to investigate the association between simulation design characteristics and educational practices associated with student satisfaction and self-confidence in learning (SCLS), an important simulation outcome. By identifying the specific elements that enhance SCLS amid the demands for diverse simulation scenarios and difficulties, we aim to present foundational data for implementing diverse types of SBL.

In this context, we formulated the following study questions: (1) What are the SCLS, Simulation Design Scale (SDS), and Educational Practice in Simulation (EPSS) scores among nursing students? (2) How are the SCLS, SDS, and EPSS correlated? and (3) What are the predictors of SCLS?

## 2. Materials and Methods

### 2.1. Study Design

This study design was an analytical cross-sectional study.

### 2.2. Participants

Fourth-year nursing students at Eulji University in the city of Seongnam, South Korea, who were taking a medical–surgical nursing simulation course and voluntarily provided informed consent to participate in the study, were enrolled. The sample size was determined using the G*power version 3.1.9.6 (Heinrich-Heine-Universität, Düsseldorf, Düsseldorf, Germany) software [16]. For linear multiple regression to achieve a significance level (α) of 0.05, effect size (f^2^) of 0.27 [17], and power (1 − β) of 0.90, number of predictors 5, the sample size was calculated to be 67. A total of 71 participants were enrolled, and 71 questionnaires were included in the final analysis.

### 2.3. Simulation Structure

In this study, the simulation course was run using the scenario of a patient with dyspnea in an adult simulation practicum course. The specific topic of the scenario was “care of a patient with chronic obstructive pulmonary disorder (COPD) showing shortness of breath”. The simulation session consisted of an orientation, self-learning, skill practice, performance evaluation, engagement in the simulation scenario, and debriefing. More specifically, the self-learning portion concerned the indications and methods for oxygen therapy and assessment and interventions for a patient with dyspnea. Nursing skills included taking vital signs, pulse oximetry, using an electrocardiogram (ECG) monitor, applying oxygen via a nasal cannula, and administering subcutaneous injections. Debriefing consisted of descriptions, analyses, and applications.

### 2.4. Scenario Overview for Patient with Dyspnea

#### 2.4.1. Topic

Nursing care for chronic obstructive pulmonary disease patients complaining of shortness of breath.

#### 2.4.2. Learning Objectives

(1)The physical and psychological state of the subject complaining of shortness of breath can be assessed.(2)Nursing interventions required for chronic obstructive pulmonary disease patients complaining of dyspnea can be applied.(3)Can communicate effectively and share roles within the health care team.(4)Can explain the mechanism of the nursing problem complained of by the subject and solve the nursing problem based on priority.

#### 2.4.3. Operating Component

(1)Simulator: High-fidelity SimMan(2)Orientation time: 20 min(3)Scenario running time: 20 min(4)Debriefing time: 30 min(5)Primary medical Diagnosis: Acute exacerbation of COPD (bronchopneumonia)(6)pre-learning
Subject education for nebulization using SVN (small volume nebulizer)Humidification application using LVN (large volume nebulizer)ABGA (Arterial Blood Gas Analysis) readingChest physiotherapy (with palm cup)Nasopharyngeal (oropharyngeal) suctionAmbu bagging, preparing intubation

### 2.5. Research Ethics

This study used an online survey of nursing students to assess their simulation course and thus had no risk to the students. The instructor informed the students of the purpose and method of the study, as well as the fact that the survey was anonymous, and that participation would not be graded. The collected data were only accessible to the researcher, and the participants were also informed that the data would only be used for research purposes, that they had the freedom to withdraw from the study at any time, and that they had the right to request their data be excluded from the analysis.

### 2.6. Tools

#### 2.6.1. Student Satisfaction and Self-Confidence in Learning (SCLS)

The SCLS is a scale developed by the NLN [18] to assess learning satisfaction and confidence during simulations. This is a 13-item scale with two subscales: satisfaction with current learning (five items) and self-confidence in learning (eight items). Each item was rated on a five-point scale from 1 (“strongly disagree” and 5 “strongly agree”), with a higher score indicating greater overall satisfaction with, and confidence in, simulation education. We obtained permission to use the tool from the NLN and the Korean-validated version of the tool [19]. Cronbach’s alpha for satisfaction with the current learning subscale was 0.94 in the studies by Jeffries and Rizzolo [20], 0.71 in the study by Hur et al. [19], and 0.97 in this study. In Jeffries and Rizzolo’s [20] study, Cronbach’s alpha for the self-confidence in learning subscale was 0.87 in the study by Jeffries and Rizzolo [19], 0.70 in the study by Hur et al. [19], and 0.87, respectively. In our study, Cronbach’s alpha for the SCLS was 0.93.

#### 2.6.2. Simulation Design Scale-Student Version (SDS)

The SDS is an instrument developed by the NLN [18] to assess whether simulation design features are well implemented in the simulation. It is a 20-item tool comprising five subscales: Objectives and Information (five items), support (four items), problem-solving (five items), Feedback/Guided Reflection (four items), and fidelity (two items). Each item was rated on a five-point scale from 1 (strongly disagree) to 5 (strongly agree), with a higher score indicating an increased recognition of design features in the simulation. We obtained permission from the NLN to use the tool and used the Korean-validated version of the tool [19]. The Cronbach’s alpha was reported as 0.92 in the study by Jeffries and Rizzolo [20], 0.88 in the study by Hur et al. [19], and 0.96 in this study.

#### 2.6.3. Educational Practices in Simulation Scale (EPSS)

The EPSS was developed by the NLN [18] to assess whether educational best practices were used in the simulation. It is a 16-item tool with four subscales: active learning (10 items), collaboration (2 items), diverse ways of learning (2 items), and high expectations (2 items). Each item was rated on a five-point scale from 1 (strongly disagree) to 5 (strongly agree), with a higher score indicating increased recognition of the use of educational best practices in the simulation and a better educational situation. We obtained permission from the NLN to use the tool, and we used the Korean-validated version of the tool [19]. The Cronbach’s alpha was reported as 0.86 in the study by Jefrries and Rizzolo [20], 0.85 in the study by Hur et al. [19], and 0.94 in this study.

### 2.7. Data Collection

Data were collected from 1 October 2019 to 11 October 2019. A recruitment announcement containing the QR code and URL for the Google online survey was posted on a bulletin board in front of the simulation room. The participants were informed about the purpose and method of the study, their rights and confidentiality, and how to participate. Participants accessed the survey via a QR code or URL, and those who clicked “I agree” after reading the study information were allowed to proceed with the survey. The participants’ general characteristics and SCLS, SDS, and EPSS scores were collected.

### 2.8. Statistical Analysis

The collected data were analyzed using IBM SPSS Statistics 25.0 (IBM, New York, NY, USA). Nursing students’ characteristics were analyzed using means and standard deviations, frequencies and percentages, and minimum and maximum values. The SCLS, SDS, and EPSS scores were presented as mean with standard deviation and minimum with maximum. The normality test of the variables was performed with the Kolmogorov–Smirnov test. Differences in the SCLS score according to participant characteristics were analyzed using independent *t*-tests. Correlations among nursing students’ SCLS, SDS, and EPSS scores were analyzed using Pearson’s correlation coefficients. The effects of student characteristics and SDS and EPSS scores on SCLS scores were analyzed using stepwise multiple linear regression analysis. Statistical significance was set at *p* < 0.05.

## 3. Results

### 3.1. Characteristics of the Participants

The mean age of the participants was 23.24 ± 1.79 years (range: 21~29), and most of the students were between ages 21–24 years (*n* = 57, 80.3%). Sixty-one of the students were female (88.7%). Eight students (11.3%) had prior simulation experience, while 63 (88.7%) did not (Table 1).

### 3.2. SCLS, SDS, and EPSS

The mean total SCLS score was 56.31 ± 7.26 (range: 37~65), with a mean score of 22.20 ± 3.72 (range: 15~25) for the Satisfaction with current learning subscale and 34.11 ± 4.36 (range: 23~40) for the Self-confidence in learning subscale. The mean SDS score was 86.82 ± 10.19 (range: 64~100), and the mean EPSS score was 70.87 ± 7.66 (range: 53~80) (Table 2).

### 3.3. Difference in SCLS by Characteristics of the Participants

There were no significant differences in the SCLS scores according to the participants’ characteristics (Table 3).

### 3.4. Correlation among SCLS, SDS, and EPSS

SCLS positively correlated with SDS (r = 0.74, *p* < 0.001) and EPSS (r = 0.75, *p* < 0.001) (Table 4).

### 3.5. Predictors of SCLS

A stepwise multiple regression analysis was performed to identify the predictors of SCLS in nursing students (Table 5). Categorical variables, namely, sex and prior simulation experience, were dummy-coded, and age, SDS, and EPSS were entered as continuous variables. When generating the model, variables were excluded with reference to a *p*-value of 0.10, and variables were selected with reference to a *p*-value of 0.05. In the SCLS model, tolerance among the independent variables was above the cutoff of 0.1, and the variance inflation factor was below the cutoff of 10, confirming the absence of multicollinearity. The SCLS model showed that SCLS increased with increasing EPSS (β = 0.43, t = 3.01, *p* = 0.004) and SDS (β = 0.38, t = 2.62, *p* = 0.011). SDS and EPSS explained 58.7% of the variance in SCLS (F = 50.83, *p* < 0.001).

## 4. Discussion

In this study, we implemented a simulation course for nursing students and examined the relationship between educational practices in simulation, simulation design characteristics, and the outcomes of simulation (SCLS) in accordance with Jeffries’s [10] nursing education simulation model. The SCLS consists of two subscales: Satisfaction with Current Learning and Self-confidence in Learning.

The scores for both subscales of the SCLS were high, above 4.2 out of 5, consistent with previous findings showing high satisfaction with simulation education overall [21,22]. The Satisfaction with Current Learning subscale consisted of items on whether the teaching method was helpful and effective, whether the instructor enjoyed teaching the simulation course, whether the teaching materials were motivating and helpful, and whether the students liked the instructor’s teaching style. Our results indicated that students were generally highly satisfied with all these components.

The Self-confidence in Learning subscale scores were also high in our study. Confidence scores were inconsistent across previous studies [22,23]. Although practicing in a simulated environment that closely resembles real-life situations can enhance students’ confidence, facing a real situation that proves to be more challenging than expected may actually lower their confidence. These results emphasize the significance of simulation design rather than simply offering simulations per se.

In our study, EPSS and SDS were identified as predictors of SCLS among nursing students, with these two factors explaining 58.7% of the variance. The positive correlation between EPSS, SDS, and SCLS among nursing students suggests that educational outcomes improve with more desirable educational practices and simulation designs, supporting previous findings [22]. Moreover, the high percentage of explained variance (58.7%) indicates a strong association of educational practices and design characteristics with educational outcomes. Further, it accentuates the significance of these components in learning performance. The results are discussed in detail below.

The SDS assesses whether the simulation features are well implemented in the actual simulation and comprises the following subscales: Objectives and Information, support, problem-solving, feedback/guided reflection, and fidelity (realism). Our finding that the SDS predicted SCLS could be interpreted in terms of these subscales as follows:

The Objectives and Information subscale pertains to pre-briefing, whether the direction of the simulation and adequate information were presented at the beginning of the simulation to motivate learning, and whether the student had a clear understanding of the purpose and objectives of stimulation. In our study, the objectives and information scores were high at 4.2, partially supporting a previous report in which the objectives and information scores were relatively low at 3.9 [22]. Providing learning objectives and information related to the types of activities performed before running the simulation and simulation-based clinical practicum in nursing education can be designed with varying levels of difficulty overall, as well as for each simulation component. In other words, setting clear objectives and providing relevant information according to a specific simulation design are associated with SCLS.

The Support subscale pertains to whether timely support has been provided, whether students recognize that they require assistance, and whether they received support from the faculty during the learning process. The SCLS increased with increasing support. Although support generally refers to support from the instructor, it may also encompass mutual support (i.e., support shared within the team). Studies have demonstrated that mutual support can lead to temporary modifications and the redistribution of workflow responsibilities in response to team members who cannot achieve their work goals or expected outcomes alone [24]. As mutual support has been found to improve adaptability and promote fluid adjustments to enhance the status quo [25], this result suggests that organized and seamless team activities are linked to SCLS.

The problem-solving subscale pertains to whether students were encouraged to explore all possibilities during a simulation scenario and were given an opportunity to set patient goals and prioritize their nursing process. During the simulation, students are presented with a situation and must identify and prioritize problems and develop solutions. Previous research has demonstrated that nursing students who underwent simulation training significantly improved their problem-solving skills [26]. Therefore, it is evident that achieving a learning outcome—solving given problems—positively impacts both satisfaction and confidence.

The Feedback and guided reflection subscale pertained to debriefing. In our study, feedback and guided reflection predicted SCLS, which supports previous findings that intentional practice and video debriefing effectively promote students’ acquisition of nursing and self-evaluation skills in simulations [27]. Reflective thinking about one’s experiences is critical for knowledge construction [28], and feedback is the most vital component of effective simulations [29]. Additionally, it has been suggested that students sharing feedback with their instructors during debriefing can enhance learning outcomes [30], indicating that reflective thinking and appropriate feedback from peers and instructors during debriefing can enhance SCLS through simulation.

The Fidelity (realism) subscale assessed whether the scenario was similar to a real-life situation and whether the simulation resembled the clinical setting. In our study, we found that students who perceived a high level of fidelity demonstrated high satisfaction and confidence, which aligns with previous research [31], indicating that simulation learning that closely resembles real-life situations can promote a sense of safety and satisfaction with clinical experiences. Since fidelity is affected by various factors related to the environment, tools, resources, and participant-related factors [32], providing adequate resources is essential for improving fidelity.

The EPSS is another predictor of SCLS. The EPSS represents the use of educational best practices in simulation, consistent with previous findings indicating a significant positive correlation between multifaceted educational practices and SCLS [33]. The EPSS comprises active learning, collaboration, and diverse learning subscales. In our study, the EPSS score ranged from 4.3–4.6 out of 5, comparable to the score of 4.2–4.5 for PESS in a previous study that developed and evaluated the appropriateness of an emergency management simulation lab curriculum for nursing students [13]. The influence of the EPSS on the SCLS can be interpreted in terms of its subscales as follows:

The first EPSS subscale includes active learning. This subscale deals with whether students were given an opportunity to discuss the ideas and concepts learned during the simulation with other students and instructors and whether they had an opportunity to reflect deeply on their pre-existing opinions during the debriefing. In the present study, active learning was associated with SCLS. As the planning of regular and structured learning experiences has been reported to foster clinical judgment even with the same scenarios and debriefing criteria, which leads to enhanced performance in simulation [34], it is crucial for students to actively engage in discussing the concepts learned during the simulation process with instructors and peers.

The Collaboration subscale assesses student collaboration, using items such as “During the simulation, my peers and I had to work on the clinical situation together.” The fact that SCLS increased with increasing collaboration scores suggests that students’ SCLS increases when they are satisfied with the team activities, as the simulation is a learning modality that emphasizes the importance of collaboration [35]. Previous studies have stressed the importance of collaboration and teamwork [36], and it is likely that good collaboration in an environment that closely resembles an actual clinical setting leads to enhanced satisfaction and confidence. Effective communication is a key component of collaboration, and as communication entails clarification and follow-up to ensure that team members have a shared understanding and perception [24], communication for collaboration can also be viewed as a factor associated with teamwork, satisfaction, and confidence.

The Diverse Ways of Learning subscales assessed whether various educational materials were provided in the simulation. The strong association between diverse learning methods and SCLS observed in our study aligns with previous findings that implementing diverse learning styles in nursing simulations can lead to high student satisfaction and confidence, even in the same simulation scenario [37]. It has been proposed that videos, websites, role-playing, and small group practices can be employed in addition to lectures and printouts for simulation education [38] so diverse learning methods can be utilized in simulation.

The High Expectations subscale assesses whether the objectives of the simulation were clear and easy to understand and whether the instructor communicated the objectives and expectations to be fulfilled during the simulation. Since gaining clinical experience only through simulation can cause stress and cognitive load to students [39], clearly sharing appropriate goals and implications is crucial. It is also important to establish high expectations and help students achieve them.

This study explored the major features of simulation education from students’ perspectives. Our findings underscore the importance of improving the SDS and EPSS to boost students’ SCLS, which is necessary to advance the overall aspects of the simulation.

This study’s main limitation is that it was conducted at one university, so it is necessary to be cautious about generalizing. It is suggested that it be applied to more diverse colleges and grades and various scenarios in the future.

## 5. Conclusions

In this study, we implemented a simulation practicum course for nursing students and examined the associations between the EPSS, SDS, and outcomes of simulation (SCLS) based on Jeffries’s nursing education simulation model. We explored the major features of the simulation and found that improving the SDS and EPSS was crucial for advancing the overall aspects of the simulations. Through the results of this study, in order to improve learning satisfaction and learning confidence of nursing students in simulation classes, it was found that consideration of simulation design and practice considering educational factors is necessary. Therefore, for this purpose, it will be necessary for instructors to develop simulation design competency in order to consider the corresponding element in simulation education.

## Figures and Tables

**Table 1 healthcare-11-01060-t001:** Participant characteristics (*n* = 71).

Characteristics		N (%)	M ± SD	Range
Age (year)	21–24	57 (80.3)	23.24 ± 1.79	21~29
	25–29	14 (19.7)		
Sex	Male	10 (14.1)		
	Female	61 (85.9)		
Simulation learning experience	No	63 (88.7)		
	Yes	8 (11.3)		

Note. N, frequency; M, mean; SD, standard deviation; min, minimum; max, maximum.

**Table 2 healthcare-11-01060-t002:** Descriptive statistics of variables.

Variables	Items	M ± SD	Range
Student satisfaction and self-confidence in learning	13	56.31 ± 7.26	37~65
Simulation design scale	20	86.82 ± 10.19	64~100
Educational practices in simulation scale	16	70.87 ± 7.66	53~80

Note. M, mean; SD, standard deviation; min, minimum; max, maximum.

**Table 3 healthcare-11-01060-t003:** Differences in student satisfaction with and self-confidence in learning according to participant characteristics (*n* = 71).

Characteristics		N (%)	M ± SD	t (*p*)
Age (year)	21–24	57 (80.3)	56.32 ± 7.18	0.01 (0.989)
	25–29	14 (19.7)	56.29 ± 7.83	
Sex	Male	10 (14.1)	57.70 ± 5.31	0.65 (0.517)
	Female	61 (85.9)	56.08 ± 7.54	
Simulation learning experience	No	63 (88.7)	56.41 ± 7.54	0.33 (0.740)
	Yes	8 (11.3)	55.50 ± 4.66	

Note. N: frequency; M: mean; SD: standard deviation.

**Table 4 healthcare-11-01060-t004:** Correlations among variables.

Variables	Student Satisfaction and Self-Confidence in Learning
	r (*p*)
Student satisfaction and self-confidence in learning	1
Simulation design scale	0.74 (<0.001)
Educational practices in simulation scale	0.75 (<0.001)

**Table 5 healthcare-11-01060-t005:** Factors associated with Student Satisfaction and Self-confidence in Learning (*n* = 71).

Variables	B	SE	β	t (*p*)
Constant	4.18	5.24		0.80 (0.428)
Simulation Design Scale	0.27	0.10	0.38	2.62 (0.011)
Educational Practices in Simulation Scale	0.41	0.14	0.43	3.01 (0.004)
F (*p*)	50.83 (<0.001)			
Adjusted R^2^	0.587			
Tolerance	0.29			
Variance inflation factor	3.47			
Durbin–Watson	1.80			

Note. B: Unstandardized Regression Coefficient, SE: standard error, β: standardized Regression Coefficient.

## Data Availability

Data sharing is not applicable.

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
