# Peer review of "Factors Associated with Student Satisfaction and Self-Confidence in Simulation Learning among Nursing Students in Korea"

_healthcare, 2023, doi:10.3390/healthcare11081060_

Round 1
Reviewer 1 Report
Dear Authors,
the manuscript is written correctly, however, I have some doubts regarding the methodology.
Please, read the attached comments.
Abstract
Nothing is written about needs to perform this study. (purpose of the study).
It is not necessary to put information about exact university in the abstract. Actually, to form city S and University E look awkward, please, resign from this information. It is enough if you mention that study was performed on 4 year students.
As well, I suggest to resign from information about statistical software and type of statistics used, the statistical analysis section in the manuscript is the place where that info should be located. The description of the results are modes please, expand.
Introduction
Is well written.
Material and Methods
Cross-sectional study it is an analytical study because its main aim about assessing the relation between factor and an outcome. I would remove section 2.1 Study Design because, aims are explained in the introduction, secondly the used methods directly explain which type of the study it is.
Why names of the University and city are blinded?
Why authors perform multivariate regression if they calculated necessary sample size based on correlation?
According to regression assumption, and number of used independent factors, the sample size should be bigger.
Why students from 4 year were took into the consideration only?
Table 2.
I don’t think that titulus sign is appropriate between min. max.
Table 3. What is the reason for splitting age into the age groups.?
Conclusion should correspond to the aims of the study while they are not. Please, paraphrase the conclusions into the form where they will show answers for research questions. Same in the abstract.
References should be more standardized. [CrossRef] {PubMed] such notes are not necessary.
Good luck.
Author Response
Response to Reviewer 1 Comments
Comments and Suggestions for Authors
the manuscript is written correctly, however, I have some doubts regarding the methodology. Please, read the attached comments.
Authors’ Response: We appreciate the time and effort that you and the reviewers have put in providing valuable feedback and insightful comments, which improved our manuscript. We have carefully considered each comment and updated the manuscript, as required.
We have marked the revisions made to the manuscript red font.
Point 1: Abstract: Nothing is written about needs to perform this study. (purpose of the study). It is not necessary to put information about exact university in the abstract. Actually, to form city S and University E look awkward, please, resign from this information. It is enough if you mention that study was performed on 4year students. As well, I suggest to resign from information about statistical software and type of statistics used, the statistical analysis section in the manuscript is the place where that info should be located. The description of the results are modes please, expand.
Response 1: The purpose, subjects, and results were revised in the abstract.
Abstract: This study aimed to examine the relationships between student satisfaction and self-confidence in learning (SCLS), the simulation design scale (SDS), and educational practices in simulation (EPSS) and to identify the influencing factors on SCLS in nursing students undergoing simulation learning. Of the fourth-year nursing students, 71 who were taking a medical-surgical nursing simulation course and voluntarily provided informed consent to participate in the study were enrolled. Data on SCLS, SDS, and EPSS were collected via an online survey after the simulation from October 1, 2019, to October 11, 2019. The mean SCLS score was 56.31 ± 7.26, the mean SDS score was 86.82 ± 10.19(range:64~100), and the mean EPSS score was 70.87 ± 7.66(range:53~80). SCLS was positively correlated with SDS (r=0.74, p <0.001) and EPSS (r=0.75, p<0.001). The regression model for SCLS in nursing students revealed that SCLS increased with increasing EPSS and SDS, and that SDS and EPSS explained 58.7% of the variance in SCLS (F=50.83, p<0.001). Therefore, to improve the learning satisfaction and learning confidence of nursing students in simulation classes, it is necessary to consider simulation design and practice considering educational factors.
Point 2: Material and Methods: Cross-sectional study it is an analytical study because its main aim about assessing the relation between factor and an outcome. I would remove section 2.1 Study Design because, aims are explained in the introduction, secondly the used methods directly explain which type of the study it is.
Response 2: Thank you for your valuable feedback. Redundant purposes were deleted and only the research design was described.
2.1. Study Design
This study design was an analytical cross-sectional study.
Point 3: Material and Methods: Why names of the University and city are blinded?
Response 3: We revealed the name of the university and city.
2.2. Participants
Fourth-year nursing students at Eulji University in the city of Seongnam, South Korea, who were taking a medical-surgical nursing simulation course and voluntarily provided informed consent to participate in the study, were enrolled.
Point 4: Material and Methods: Why authors perform multivariate regression if they calculated necessary sample size based on correlation?
Response 4: We appreciate your thoughtful comment. Based on linear multiple regression analysis, the number of samples was recalculated and revised.
2.2. Participants
Fourth-year nursing students at Eulji University in the city of Seongnam, South Korea, who were taking a medical-surgical nursing simulation course and voluntarily provided informed consent to participate in the study, were enrolled. The sample size was determined using the G*power version 3.1.9.6 software [16]. For linear multiple regression to achieve a significance level (α) of 0.05, effect size (f2) of 0.27 [17], and power (1-β) of 0.90, number of predictors 5, the sample size was calculated to be 67. A total of 71 participants were enrolled, and 71 questionnaires were included in the final analysis.
Point 5: Material and Methods: According to regression assumption, and number of used independent factors, the sample size should be bigger.
Response 5: The appropriate number of samples required for the multiple regression analysis of this study was 67, and all 71 fourth-year students participated in this study. When the power was recalculated as a result of this study, it was 0.999.
Citation
Knofczynski, G. T., & Mundfrom, D. (2008). Sample sizes when using multiple linear regression for prediction. Educational and Psychological Measurement, 68(3), 431-442. https://doi.org/10.1177/0013164407310131
Point 6: Material and Methods: Why students from 4 year were took into the consideration only?
Response 6: Nursing departments in Korea vary slightly from university to university, but in general, the curriculum consists of liberal arts for the first year, basic majors and major electives for the second year, and mandatory majors and clinical practice for the third and fourth years. The 4th grade was the last grade, and the 4th grade was targeted because it conducted simulation practice applying clinical situations.
Point 7: Table 2. I don’t think that titulus sign is appropriate between min. max.
Response 7: Thank you for your valuable feedback. The titulus sign has been corrected.
Point 8: Table 3. What is the reason for splitting age into the age groups.?
Response 8: In the case of normal cases, 4th graders in Korea are 23 years old, and the average age is 23.24 years old, which is divided into under 24 years old and over 25 years old. I had an expectation that older students aged 25 or older would be able to perform the nursing practice well in a simulation situation because they had a lot of life experience, but there was no difference.
Point 9: Conclusion should correspond to the aims of the study while they are not. Please, paraphrase the conclusions into the form where they will show answers for research questions. Same in the abstract.
Response 9: Thank you for your valuable feedback. We added the following to the abstract and conclusion sections.
Abstract: This study aimed to examine the relationships between student satisfaction and self-confidence in learning (SCLS), the simulation design scale (SDS), and educational practices in simulation (EPSS) and to identify the influencing factors on SCLS in nursing students undergoing simulation learning. Of the fourth-year nursing students, 71 who were taking a medical-surgical nursing simulation course and voluntarily provided informed consent to participate in the study were enrolled. Data on SCLS, SDS, and EPSS were collected via an online survey after the simulation from October 1, 2019, to October 11, 2019. The mean SCLS score was 56.31 ± 7.26, the mean SDS score was 86.82 ± 10.19(range:64~100), and the mean EPSS score was 70.87 ± 7.66(range:53~80). SCLS was positively correlated with SDS (r=0.74, p <0.001) and EPSS (r=0.75, p<0.001). The regression model for SCLS in nursing students revealed that SCLS increased with increasing EPSS and SDS, and that SDS and EPSS explained 58.7% of the variance in SCLS (F=50.83, p<0.001). Therefore, to improve the learning satisfaction and learning confidence of nursing students in simulation classes, it is necessary to consider simulation design and practice considering educational factors.
- Conclusion
In this study, we implemented a simulation practicum course for nursing students and examined the associations between the EPSS, SDS, and outcomes of simulation (SCLS) based on Jeffries’s nursing education simulation model. We explored the major features of the simulation and found that improving the SDS and EPSS was crucial for advancing the overall aspects of the simulations. Through the results of this study, in order to improve learning satisfaction and learning confidence of nursing students in simulation classes, it was found that consideration of simulation design and practice considering educational factors are necessary. Therefore, for this purpose, it will be necessary for instructors to develop simulation design competency in order to consider the corresponding element in simulation education.
Point 10: References should be more standardized. [CrossRef] {PubMed] such notes are not necessary.
Response 10: Thank you for your valuable feedback. We removed references such as [CrossRef] [PubMed] from the reference list. The format of the reference list has been modified to be consistent.
Reviewer 2 Report
Thank you very much for the opportunity to review this very interesting article.
Title: the word simulation should appear
Abstract: improve the wording of the general objective.
Introduction: Line (32) Avoid using alternative, simulation is complementary to clinical practicum. The rest of the text is correct; the study is justified very well.
I have problems between the questions and the objectives, it should be clarified. In the questions the characteristics of the students appear but not in the objective, it should align correctly.
Materials and Methods.
Participants: it is necessary to know the total population, not only the sample. Briefly describe the training in Korea for comparison. Are the fourth-year students in their last year?
I suggest creating a Context section, to explain the development of the clinical simulation. It would be nice to have the simulation session explained in more detail: high-fidelity mannequins or simulated patients, simulation objectives, simulation phases with dedicated times and debriefing model.
Statistical analysis: was the Kolmogorov-Smirnov test in R (Lilliefors) revised to test for normality? It would be indicated to do so to determine the subsequent statistical tests.
Results:
• According to information Table 1 the students did not have experience in simulation? Was this the first?
• Improve format of Table 2, too much information.
Discussion: congratulations to the authors for the discussion. A section of limitations is lacking.
Author Response
Response to Reviewer 2 Comments
Comments and Suggestions for Authors
Thank you very much for the opportunity to review this very interesting article.
Authors’ Response: We appreciate the time and effort that you and the reviewers have put in providing valuable feedback and insightful comments, which considerably improved our manuscript. We have carefully considered each comment and made changes to the manuscript, as required. We have marked in red font the revisions we made in the manuscript.
Point 1: Title: the word simulation should appear
Response 1: We would like to convey our deepest gratitude for your review on our study. We revised title as follows.
Factors Associated with Student Satisfaction and Self-Confidence in Simulation Learning among Nursing Students in Korea
Point 2: Abstract: improve the wording of the general objective.
Response 2: We appreciate your critical comment. The purpose of the abstract was modified as follows.
This study aimed to examine the relationships between student satisfaction and self-confidence in learning (SCLS), the simulation design scale (SDS), and educational practices in simulation (EPSS) and to identify the influencing factors on SCLS in nursing students undergoing simulation learning.
Point 3: Introduction: Line (32) Avoid using alternative, simulation is complementary to clinical practicum. The rest of the text is correct; the study is justified very well.
Response 3: Thank you for your thoughtful comment. We deleted the expression alternative.
- Introduction
Simulation-based learning (SBL) in nursing provides a safe environment for clinical practicum in which students can explore nursing scenarios and gain clinical exposure [1] and is utilized in various forms in nursing education [2]. The World Health Organization (WHO) recommends using simulation in education, as it helps students acquire skills and attain improved learning outcomes [3]. In particular, the importance of SBL is further highlighted as an actual clinical practicum owing to the increased barriers to clinical practicum amid a climate focusing on patient safety, human rights, and an infectious disease crisis, namely the coronavirus disease 2019 (COVID-19) pandemic. SBL has been documented to enhance clinical judgment, problem-solving skills, and critical thinking [4] and improve learning satisfaction, confidence, and self-efficacy [5].
Point 4: I have problems between the questions and the objectives, it should be clarified. In the questions the characteristics of the students appear but not in the objective, it should align correctly.
Response 4: Thank you for your valuable feedback. We modified the question to align with the objective.
In this context, we formulated the following study questions: (1) What are the SCLS, Simulation Design Scale (SDS), and Educational Practice in Simulation (EPSS) scores among nursing students? 2) How are the SCLS, SDS, and EPSS correlated? and 3) What are the predictors of SCLS?
Point 5: Materials and Methods: Participants: it is necessary to know the total population, not only the sample. Briefly describe the training in Korea for comparison. Are the fourth-year students in their last year?
Response 5: The total number of 4th grade students was 71. The department of nursing in Korea is slightly different for each university, but generally, the curriculum consists of liberal arts for the first year, basic major and major electives for the second year, and compulsory major and clinical practice for the third and fourth years. The 4th year is the final year and conducts simulation practice applying clinical situations. The total number of students from 1st to 4th grade is about 280. Among them, this study targeted 4th-grade students.
Point 6: Materials and Methods: I suggest creating a Context section, to explain the development of the clinical simulation. It would be nice to have the simulation session explained in more detail: high-fidelity mannequins or simulated patients, simulation objectives, simulation phases with dedicated times and debriefing model.
Response 6: Thanks for pointing out this valuable point. We added the following contents that the simulation session explained in more detail: high-fidelity mannequins or simulated patients, simulation objectives, simulation phases with dedicated times, and a debriefing model.
2.4. Scenario Overview for Patient with Dyspnea
2.4.1. Topic
Nursing care for chronic obstructive pulmonary disease patients complaining of shortness of breath
2.4.2. Learning Objectives
1) The physical and psychological state of the subject complaining of shortness of breath can be assessed.
2) Nursing interventions required for chronic obstructive pulmonary disease patients complaining of dyspnea can be applied.
3) Can communicate effectively and share roles within the health care team.
4) Can explain the mechanism of the nursing problem complained by the subject and solve the nursing problem based on priority.
2.4.3. Operating Component
1) Simulator: High-fidelity SimMan
2) Orientation time: 20min
3) Scenario running time: 20min
4) Debriefing time: 30min
5) Primary medical Diagnosis: Acute exacerbation of COPD (bronchopneumonia)
6) pre-learning
⦁ Subject education for nebulization using SVN (small volume nebulizer)
⦁ Humidification application using LVN (large volume nebulizer)
⦁ ABGA (Arterial Blood Gas Analysis) reading
⦁ Chest physiotherapy (with palm cup)
⦁ Nasopharyngeal (oropharyngeal) suction
⦁ Ambu bagging, preparing intubation
Point 7: Materials and Methods: Statistical analysis: was the Kolmogorov-Smirnov test in R (Lilliefors) revised to test for normality? It would be indicated to do so to determine the subsequent statistical tests.
Response 7: A normality test was added to the data analysis method.
2.7. Statistical Analysis
The collected data were analyzed using IBM SPSS Statistics 25.0. Nursing students’ characteristics were analyzed using means and standard deviations, frequencies and percentages, and minimum and maximum values. The SCLS, SDS, and EPSS scores were presented as mean with standard deviation and minimum with maximum. The normality test of the variables was performed with the Kolmogorov-Smirnov test.
Point 8: Results: According to information Table 1 the students did not have experience in simulation? Was this the first?
Response 8: Thanks for pointing out this valuable point. Most of students experience simulation learning for the first time because they do clinical practice until the 3rd year and run simulation classes in the 4th year. We think that some students responded that they had experience in simulation education because they supported for simulation classes as working students in their 3rd year.
Point 9: Results: Improve format of Table 2, too much information.
Response 9: Thanks for pointing out this valuable point. Sub-areas of variables were deleted and concisely organized.
Table 2. Descriptive statistics of variables.
|
Variables |
Items |
M ± SD |
Range |
|
Student satisfaction and self-confidence in learning |
13 |
56.31 ± 7.26 |
37~65 |
|
Simulation design scale |
20 |
86.82 ± 10.19 |
64~100 |
|
Educational practices in simulation scale |
16 |
70.87 ± 7.66 |
53~80 |
Point 10: Discussion: congratulations to the authors for the discussion. A section of limitations is lacking.
Response 10: Thank you for your valuable feedback. We added the limitations in the discussion section as follows.
This study explored the major features of simulation education from students’ perspectives. Our findings underscore the importance of improving the SDS and EPSS to boost students’ SCLS, which is necessary to advance the overall aspects of the simulation.
This study's main limitation is that it was conducted at one university, so it is necessary to be cautious about generalizing. It is suggested that it be applied to more diverse colleges and grades and various scenarios in the future.